# miR-497 Regulates *LATS1* through the *PPARG* Pathway to Participate in Fatty Acid Synthesis in Bovine Mammary Epithelial Cells

**DOI:** 10.3390/genes14061224

**Published:** 2023-06-05

**Authors:** Shuangfeng Chu, Yi Yang, Mudasir Nazar, Zhi Chen, Zhangping Yang

**Affiliations:** 1College of Animal Science and Technology, Yangzhou University, Yangzhou 225009, China; dx120200142@yzu.edu.cn (S.C.); drmudasirnazar457@gmail.com (M.N.); zhichen@yzu.edu.cn (Z.C.); 2Joint International Research Laboratory of Agriculture & Agri-Product Safety, Ministry of Education, Yangzhou University, Yangzhou 225009, China; 3College of Veterinary Medicine, Yangzhou University, Yangzhou 225009, China; yangyi@yzu.edu.cn

**Keywords:** miR-497, *LATS1*, bovine mammary epithelial cells, milk fat metabolism

## Abstract

Nutrient metabolism is required to maintain energy balance in animal organisms, and fatty acids play an irreplaceable role in fat metabolism. In this study, microRNA sequencing was performed on mammary gland tissues collected from cows during early, peak, and late lactation to determine miRNA expression profiles. Differentially expressed miRNA (miR-497) was selected for functional studies of fatty acid substitution. Simulants of miR-497 impaired fat metabolism [triacylglycerol (TAG) and cholesterol], whereas knockdown of miR-497 promoted fat metabolism in bovine mammary epithelial cells (BMECs) in vitro. In addition, in vitro experiments on BMECs showed that miR-497 could down-regulate C16:1, C17:1, C18:1, and C20:1 as well as long-chain polyunsaturated fats. Thus, these data expand the discovery of a critical role for miR-497 in mediating adipocyte differentiation. Through bioinformatics analysis and further validation, we identified large tumor suppressor kinase 1 (*LATS1*) as a target of miR-497. siRNA-*LATS1* increased concentrations of fatty acids, TAG, and cholesterol in cells, indicating an active role of *LATS1* in milk fat metabolism. In summary, miR-497/*LATS1* can regulate the biological processes associated with TAG, cholesterol, and unsaturated fatty acid synthesis in cells, providing an experimental basis for further elucidating the mechanistic regulation of lipid metabolism in BMECs.

## 1. Introduction

Milk is rich in nutrients and gradually becomes a part of people’s daily diet. Ruminant milk is the main edible dairy product worldwide, and milk is the most important source of human dairy products. Milk protein, lactose, milk fat, vitamins, and metal elements play crucial roles in human health and nutritional regulation [1]. Milk formation involves the biosynthesis and secretion of fat, protein, and lactose [2], as well as a variety of signaling pathways, such as the *PPARG* signaling pathway [3], the JAK/STAT5 signaling pathway [4], and amino acid and glucose uptake pathways [5], which are mainly involved in the regulation between miRNAs and mRNAs, and increasing studies have shown that miRNAs are also important regulators of fat metabolism in breast tissue [6,7]. MiR-33 can regulate lipid metabolism in a variety of ways, and miR-33a/b can inhibit the expression abundance of ATP-binding cassette transporter A1 (*ABCA1*) and regulate the cellular transport of cholesterol and the biosynthesis of high-density lipoprotein [8,9], which are essential to maintain intracellular cholesterol homeostasis [10]. Studies have found that there are differences in the regulation of milk fat metabolism for different miRNAs; miR-33a and miR-103 overexpression can promote triglyceride synthesis in breast epithelial cells [11], and miR-25 and miR-27a can inhibit the expression of target genes’ *PPARs* and reduce intracellular triglyceride synthesis [6,12]. Studies in GMECs have shown that miR-135b functions through *LATS2* (a component of the Hippo signaling pathway) to regulate the synthesis of milk fat [13], and it is an important goal to study the function and regulation of Hippo in the context of milk fat synthesis [14]. Previous research results have shown that the control pairs of miR-16a and *LATS1*, as well as miR-497 and *LATS2*, are closely related to milk fat metabolism [15,16]. However, the molecular mechanisms of *LATS1* and miR-497 are specifically regulated by fatty acids, but whether the effects of miR-497 on *LATS1* and *LATS2* differ is unknown, so we further investigated their functional and regulatory relationships using molecular techniques, and biological and functional assays provided a theoretical reference for studying the genetic mechanisms of milk fat metabolism in ruminants.

Milk fat is an important component of milk, is a high-quality natural fat, is digestible up to 98%, and contains a large number of milk fat vitamins. Fat in milk consists of 98% triglycerides, small amounts of cholesterol, and free fatty acids (saturated and unsaturated fatty acids) [17]. The unsaturated fatty acids mainly include omega-3 polyunsaturated fatty acids (also known as omega-3 fatty acids), omega-6 polyunsaturated fatty acids, and conjugated linoleic acid unique to ruminants [18]. The main types of omega-3 polyunsaturated fatty acids are α-linolenic acid (ALA), octadecenoic acid (SDA), eicosatetraenoic acid (ARA), eicosapentaenoic acid (EPA), docosapentaenoic acid (DPA), and docosahexene (DHA) [19]. Unsaturated fatty acids in milk have the effects of anti-oxidation and reducing the content of triglycerides in human blood [20]. An important task of dairy cattle breeding is to increase the content of beneficial fatty acids in milk, thereby improving the economic value of the animal husbandry industry and the nutritional value of milk [21]. However, the molecular mechanism regulating the composition and content of fatty acids in milk is not clear, restricting the development of the scientific field of dairy products to a certain extent. Based on the previous results of transcriptome sequencing of miRNA in the early-, peak-, and late-lactation of dairy cows [16], the aim of this study was to bioinformatically analyze miR-497 and *LATS1* to obtain key regulatory pathways to functionally validate the molecular mechanism of milk fat metabolism for this regulatory pair and to further study the regulatory effects of miR-497 and *LATS1* in the mammary tissues and cells of dairy cows, providing a more accurate theoretical basis for molecular breeding of dairy cows.

## 2. Materials and Methods

### 2.1. Sample Collection and RNA Extraction

The experimental cattle in this study came from the farm of Yangzhou University (China). Three cows (3 years old) of similar weights were selected for mammary gland tissue collection during early lactation (7 days after parturition), peak lactation (30 days after parturition), and late lactation (315 days after parturition). Total RNA from tissue samples was extracted using Trizol reagent (Cat:15596026, Invitrogen, Carlsbad, CA, USA) and stored at −80 °C. RNA quality and quantity were assessed with an ND-1000 spectrophotometer (NanoDrop Technologies, Rockland, DE, USA).

### 2.2. Bioinformatics Analysis of Sequencing Results

Raw data were obtained by sequencing and processed for joint removal, contaminating sequences, and low-quality reads. Noncoding RNAs, such as snRNA, tRNA, and rRNA, were identified by comparison with the Rfam and Genbank databases. Clean reads were compared to the Rfam database by blastn software, and results with E-values ≤ 0.01 were extracted to annotate sequences, such as rRNA, snRNA, snoRNA, and tRNA. These comment sequences on the Rfam database were filtered and ultimately deleted. The filtered data were used for the following ratios: prediction of known miRNAs and new miRNAs [22].

### 2.3. Small RNA Length Distribution

The sequencing results contained long inserted sequences, small fragments, poly A, and lower mass sequences. Then, a clean reading was generated. Small RNA is typically 18–30 nt in length. Small RNA species were grouped according to the peak length distribution. PiRNAs peaked at 28–30 nt, siRNAs peaked at 24 nt, and miRNAs peaked at 21 or 22 nt.

### 2.4. Determination of Milk Composition

Standardized debugging of FOSS multifunctional dairy analyzer was performed to optimize the instrument state, and then spectral information of raw milk samples was collected.

### 2.5. Cell Culture and Transfection

The culture medium for BMEC includes 10% bovine serum and DMEM/F12. BMEC was cultured in 5% carbon dioxide, 37 ° C, and optimal humidity. When the cell density reached 80%, the biosynthetic reagents microRNA or siRNA were transferred (Table 1 and Appendix A), and the cells were collected for subsequent experiments after 48 h.

### 2.6. Triglyceride Assay and Cholesterol Testing

Cells (Yangzhou University, College of Veterinary Medicine) were transfected with miR-497mimic (100 nM), miR-497 inhibitor (200 nM), and siRNA-*LATS1*(100 nM) (Genepharma, Suzhou, China) using three biological replicates for each treatment. After 48 h, cells were washed with phosphate-buffered saline (PBS) (Cat: 10010023, Invitrogen, Carlsbad, CA, USA), and after removal of PBS, lysis buffer (150 μL) was added. Collected cells stood at 4 °C for 15 min. The supernatant was collected by centrifugation for cholesterol, TAG detection, and bicinchoninic acid (BCA) protein quantification. The protein concentration was finally determined using BCA values [15].

### 2.7. Determination of Intracellular Fatty Acid Content

BMECs were cultured for 24 h and treated with miR-497 MIC, inhibitor and siRNA-LATS1, and cells were collected for fatty acid extraction after 48 h. The specific method refers to the previous research in our laboratory [16]. The relative amount of fatty acids was calculated using the peak area percentage of each fatty acid, and the C19: 0 standard was used.

### 2.8. RT-qPCR and Western Blotting

Mimics and inhibitors of miR-497 were transfected into cells, and RNA and protein were extracted from transfected cells after 48 h. MiRNA expression levels were determined with the S-Poly (T) method (16). Total RNA (1 µg) was used for cDNA synthesis using a HiScript ^®^ QRT SuperMix (+ gDNA wiper) (Nanjing, China) kit for qPCR (R123-1, Vazyme, Nanjing, China). The primers were designed based on the sequences of bovine *PPARG*, *ACLY*, *CD36*, *ELOVL6*, *HSL*, and *GAPDH* in NCBI (Table 2), and real-time PCR results were analyzed using the 2^−ΔΔCt^ model.

SDS-PAGE was used to isolate proteins from cells, and they were transferred nitrocellulose membrane (Millipore, Billerica, MA, USA). Rabbit monoclonal (# 8579, Cell Signaling Technology, Shanghai, China) and monoclonal rabbit antibodies (# 8457, Cell Signaling Technology, Shanghai, China) were used, and image collection and analysis were conducted.

### 2.9. Luciferase Reporter Assay

The results showed that *LATS1* is one of the target genes of miR-497 in online analysis with TargetScan (http://www.targetscan.org/, accessed on 5 September 2021) and PicTar (http://pictar.mdc-berlin.de/, accessed on 5 September 2021). The results showed that *LATS1* was one of the target genes of miR-497. The luciferase vector containing miR-497 binding site 3′-UTR was designed and synthesized with 3′-UTR of bovine *LATS1* gene as a template (Appendix A). The 3′ UTR fragment was amplified by PCR, and the luciferase vector psi-CHECK-2 was digested with NotI and XhoI. Then, the 3′-UTR fragment was cloned into the luciferase vector of psi-CHECK-2. Luciferase reported that the plasmid was co-transfected with miR-497 mimic, and after 48 h, the luciferase activities of renila and firefly were measured by a dual GLO luciferase tester.

### 2.10. Oil Red O Staining

The BMECs were transfected with miR-497 mimic and inhibitor, which were collected after 48 h, washed three times with PBS, fixed with 10% formaldehyde for 1 h, washed with PBS three times again, stained with oil red O, observed, and photographed under a microscope after drying.

### 2.11. Statistical Analysis

The real time quantitative PCR calculation of gene expression results was performed using the 2^−ΔΔCT^ method. One-way ANOVA analysis was conducted in SPSS software, version 22.0, to analyze the differences between treatment groups. The results are displayed as the mean ± standard deviation: *p* < 0.05, *p* < 0.01.

## 3. Results

### 3.1. Sequencing Quality Analysis and Length Distribution

Analysis was performed using the double-ended 150-bp sequencing pattern of the Hiseq2500 platform. The total reading of the raw sequence was 140,023,953, and the average reading of each sample was 15,558,217. The raw data were dejoined, and the filter was sequenced with abnormal length, N, or poor quality. The total reading of the cleaning data was 132,910,778, and the average reading for each sample was 14,767,864 (Appendix A).

### 3.2. Small RNA Analysis with Typical and Unique Sequences

The common and unique sequences of small RNAs were analyzed to verify their integrity. The RNA sequences of the samples varied widely in the species, and the expression of sequences was concentrated in the general part, indicating a higher consistency of sequencing between different samples.

Genome comparison of clean data was performed with the bovine reference genome (UMD3.1) using Bowtie software. The comparison rate of nine small RNA sequencing libraries was between 85.44% and 94.27%, showing that almost all sequencing data werebovine miRNA (Appendix A).

### 3.3. Differential Expression Analysis of miRNA

The characteristics of late lactation and peak lactation were compared with those of early lactation. There were 13 miRNAs with common differential expression: bta-miR-17-5p, bta-miR-25, bta-miR-101, bta-miR-151-3p, bta-miR-199a-3p, bta-miR-15b, bta-let-7c, bta-miR-125b, bta-miR-365-3p, bta-miR-495, bta-miR-497, bta-miR-423-3p, and bta-miR-1468 (Figure 1 and Appendix A).

### 3.4. MiRNA Differential Expression Analysis

FOSS multifunctional dairy composition rapid analyzer was used to detect the milk yield and percentage content of the main components in milk during different lactation periods, and it was found that there were significant differences in all indicators (*p* < 0.05); milk yield and lactose showed first an increase and then a decreasing trend with the progress of lactation, while milk fat and milk protein showed a continuously increasing trend (Appendix A).

### 3.5. Transfection Efficiency of miRNA and siRNA

The expression level of miR-497 in miR-497 mimic group was 16 times higher than that in the NC mimic group in BEMCs (Figure 2A), and the miR-497 inhibitor group was decreased (62%) compared with the control group. The expression levels of Bos-995 (80%), Bos-43 (78%), and Bos-2446 (65%) in siRNA-*LATS1* were lower than those of siRNA-NC, so we chose Bos-43 to continue the follow-up experiment (Figure 2C).

### 3.6. miR-497 Specifically Targets LATS1 in BEMCs

The expression of *LATS1* was down-regulated (5.0-fold) after transfection of miR-497 mimic, while the expression of *LATS1* was up-regulated by 4.4 times in response to miR-497 inhibition (Figure 2B and Figure 3C). The protein expression levels of *LATS1* were significantly down-regulated after siRNA-LATS1 treatment (Figure 3D). We designed and synthesized a *LATS1* fragment containing an miR-497 target site to further demonstrate that miR-497 targets *LATS1* (Figure 3A). The luciferase report test showed that the luciferase activity constructed by wild-type 3′-UTR report decreased after miR-497 mimic transfection, but the luciferase activity reported by the mutant did not change (Figure 3B).

### 3.7. Functional Evaluation of miR-497 and LATS1 in BEMCs

#### 3.7.1. The Expression of miR-497 Reduces the Content in BEMCs

Compared with the control group, the TAG content in BMECs was decreased by 60% (*p* < 0.01) after transfection with miR-497 mimic (Figure 4A). In contrast, when miR-497 was inhibited, TAG content increased by 70% (*p* < 0.01). After transfection with mimic-miR-497, cholesterol content was decreased compared with the control, but the difference was not significant (*p* > 0.05) (Figure 4B). In contrast, when miR-497 was inhibited, the cholesterol content was increased (70%) (*p* < 0.01). In addition, the results of oil red O staining showed that overexpression of miR-497 inhibited fat droplet formation, and transfection with miR-497-inhibitor can induce fat droplet formation (Figure 4C). Regarding mRNA expression, overexpression of miR-497 increased the expression of *PPARG*, *ELOVL6*, and *GPAM* (Figure 5). In contrast, inhibition of miR-497 caused a significant decrease in fat metabolism genes, including *XDH*, *ACLY*, *PLIN3*, *CD36*, *HSL*, and *SCL27A6* (Figure 5).

#### 3.7.2. Expression of *LATS1* Increases the Level of TAG in BEMCs

The content of TAG was decreased by 82% (*p* < 0.01) compared with the control group (Figure 5A), whereas the content of cholesterol was decreased by 50% (*p* < 0.01) in the siRNA-*LATS1*-transfected cells (Figure 6B).

#### 3.7.3. Effects of miR-497 and siRNA *LATS1* on Fatty Acid Composition of Cells

The role of miRNAs in fatty acid metabolism was further determined by examining the content of fatty acid when miR-497 was overexpressed and inhibited in BMECs. Compared with the control group, miR-497 mimic treatment could significantly increase the concentrations of C12:0, C14:0, C16:0, C18:0, and C20:0 and decrease the concentrations of the saturated fatty acids C9:0, C15:0, and C17:0 after treatment. Meanwhile, the concentrations of C16:1, C17:1, C18:1, C20:1; C20:3, and C20:5 were significantly decreased, and the concentrations of C20:4 (EPA) and C22:6 (DHA) were significantly increased (Table 3); the results of treatment with miR-497 inhibitor were the opposite to those with miR-497mimic treatment of BMECs (Table 4).

Compared with the control group, the concentrations of C12:0, C14:0, C16:0, C18:0, and C20:0 were significantly increased, and the concentrations of C9:0, C15:0, and C17:0 were significantly decreased with the siRNA-*LATS1* treatment of BMECs. Meanwhile, the concentrations of C16:1, C17:1, C18:1, C20:1, C20:3, and C20:5 were significantly decreased, and the concentrations of C20:4 (EPA) and C22:6 (DHA) were significantly increased (Table 5).

#### 3.7.4. siRNA-*LATS1* Partially Abrogates (“Rescues”) the Reduction in TAG Content

The rescue experiments demonstrated that the level of TAG was significantly increased but was decreased by siRNA-*LATS1* rescue after transfection with miR-497 inhibitor (Figure 6C). SiRNA-*LATS1* rescue partially weakened the increase in TAG.

## 4. Discussion

Fatty acid content and composition in milk determine the quality of milk, and miRNAs can regulate the metabolic pathway of the mammalian breast, in turn affecting the content and composition of lipids. With the deepening of research, the important regulatory role of miRNAs in animal mammary gland development and milk fat metabolism has received increasing attention. However, it has been reported that there are significant differences in fat metabolism between breast tissues and body lipid metabolism, as well as miRNA regulation, which may be due to the expression of specific miRNAs in breast cells [23]; therefore, it is particularly important to establish miRNA expression profiles in breast tissues under different conditions with high-throughput sequencing and gene chip technology. In human studies, smRNA-Seq of breast milk lipids revealed 21 novel miRNAs, and the miRNAs present in milk exosomes play important roles in the regulation of infant development [24]. Others have found that miRNAs in breast milk are mainly derived from mammary epithelial cells rather than the maternal circulatory system [25]. In a ruminant study, sequencing analysis of expression profiles in bMECs, which were isolated from cows with extremely significant differences in milk fat percentages, revealed a total of 292 known miRNAs and 116 novel miRNAs [26]. The data from this study constitute a reference to investigate lactation mechanisms at the miRNA level. We chose miR-497 as a miRNA reagent for functional analysis to promote the synthesis of fatty acids in mammary cells. In this way, we can better understand the mechanisms of milk fat and can improve and increase the application value of milk fatty acids through genetic or practical means.

Knowing the target genes of miRNAs is essential to further clarify the regulating signal pathways, and the target gene functions of miRNAs regulating milk fat metabolism in breast tissue mainly focus on the processes of milk fat synthesis, transport, oxidation, and secretion. There are a large number of target genes related to miRNA regulation of milk fat metabolism, of which miR-135a acts on the very low-density lipoprotein receptor (*VLDLR*) gene [27], and miR-34b regulates milk fat synthesis through the target gene decapping enzyme 1A (*DCP1A*) [28], while miR-26 family target genes are involved in PI3K-Akt, the MAPK signaling pathway, and the fatty acid biosynthesis pathway [29]. Further analysis showed that a single target gene could be regulated by multiple miRNAs, such as miR-10a, miR-15b, and miR-142-5p, which may regulate milk lipid metabolism by acting on the *SLC30A4* gene [30]. In addition, miRNAs also have a synergistic regulation with target genes, and it has been found that miR-148a and miR-17-5p can synergistically act on the target genes *PGC-1* and *PPARA* in breast cells related to lipid metabolism [31]. It can be seen that miRNAs have an important regulatory effect on target genes related to fat metabolism in breast tissue, and based on the results of the previous transcriptome sequencing study, the candidate gene *LATS1* was selected in this study, and miR-497 was found to be a target by bioinformatics analysis. The results showed that *LATS1* could promote TAG synthesis by bMECs. In addition, the further results indicating that *LATS1* can promote the production of medium- and long-chain fatty acids and fat droplets in cells, providing a theoretical basis for improving the content and composition of beneficial fatty acids in milk, and it also helps to deeply dissect the regulatory network of milk fat metabolism involving miRNAs to regulate milk fat percentages and improve milk quality.

The main nutrients in animal milk are milk fat and milk protein, and these nutrients play an important role in the growth and development of young animals. Milk fat is one of the most important factors for evaluating the quality of milk because of its rich fatty acid composition and large number of fat-soluble vitamins. Factors affecting milk fat percentage include genetics, feeding management, and physiological and pathological factors, of which genetic factors are the core factors determining milk fat synthesis in dairy animals [32]. The main genes known to regulate milk lipid metabolism include hormone-sensitive triglyceride lipase (*HSL*) [33], xanthine dehydrogenase (*XDH*) [34], ATP citrate lyase (*ACLY*) [35], glycerol-3-phosphate acyltransferase (GPAM) [36], lipid droplet protein 3 (*PLIN3*) [37], cluster of differentiation 36 (*CD36*) [38], solute carrier protein family 27 member 6 (*SLC27A6*) [39], and peroxisome proliferator-activated receptors (*PPARs*) [40]. These genes have been demonstrated to play important regulatory roles in milk fat metabolism-related processes, such as de novo milk fat synthesis, fatty acid uptake and transport within the mammary gland, and oxidative breakdown of fatty acids.

Triglycerides in BMECs are mainly synthesized in the endoplasmic reticulum, fatty acyl-CoA is produced during de novo fatty acid synthesis, and exogenous uptake of fatty acids are esterified to form a skeleton of glyceryl triphosphate, which ais then generated by the action of *GPAM*. LPA is catalyzed by the *AGPAT* family to generate phosphatidic acid, which is then dephosphorylated by phospholipases to generate diacylglycerol, which is catalyzed by the *DGAT* family to generate triglycerides. *GPAM* is the rate-limiting enzyme in the first step of triglyceride synthesis, intracellular triglyceride levels are regulated by the *GPAM* enzyme [41,42], and knockdown of *GPAM* significantly reduces triglyceride synthesis and metabolism-related gene expression in bovine embryonic fibroblasts [43]. In addition, *LPIN3* belongs to the *LPIN* family, which can act as a transcriptional costimulator to regulate peroxisome proliferator-activated receptors (*PPARγ*) and PPARα and then regulate fatty acid cell differentiation and lipid metabolism, with an important impact on animal fat deposition [44]. It has been shown that overexpression of SREBP1 in goat mammary epithelial cells resulted in significant up-regulation of genes related to fatty acid synthesis, such as *ACSL1* and *ELOVL6*, and significant up-regulation of expression of triglyceride synthesis-related genes (*DGAT1* and *LPIN1*) and intracellular triglyceride content [45], consistent with the function of this study. In animal cells, fatty acid synthesis and trafficking depend on *CD36* and the fatty acid transporter *SLC27A6* [38,39]. The study showed that the expression levels of *CD36* and *SLC27A6* decreased after transfection with miR-497 mimic, consistent with the function of miR-497. The fatty acid synthesis, LCFA transport, and content of C16:1, C17:1, C18:1, C18:2, and C20:1 were enhanced, which may be responsible for the expression of *PPARG* increasing after transfection with miR-497. It has been shown that activation of *PPARG* also regulates the transcription of genes in fatty acids, such as *ACLY*, *HSL*, *LPIN1*, and *CD36* [33,38,46]. In summary, the results indicated the role of miR-497/*LATS1* in fatty acids in BMECs, and these findings may help to improve milk production and develop the pasture economy.

In recent years, *LATS1* and *LATS2* kinases have become the focus of research due to their extensive biological activities in cell cycle regulation and differentiation and their dysregulated multiple pathological outcomes. The most significant difference between *LATS1* and *LATS2* occurs at the transcriptional level. The difference in transcription factors regulating *LATS1* versus *LATS2* may represent the need to control proliferation signaling by maintaining proficient levels of LATS under both conditions. The different expression patterns of *LATS1* and *LATS2* may contribute to their likelihood of encountering the same or different binding partners [47]. This study supports such a view. There were some differences in fatty acid regulation between miR-497/*LATS2* and miR-497/*LATS2*. First, miR-497/*LATS2* was mainly involved in the regulation of five long-chain fatty acids (C16:0, C16:1, C18:0, C18:1, C18:2), while miR-497/*LATS1* was almost involved in the regulation of 17 fatty acids from C12:0 to DHA. Second, miR-497/*LATS2* is mainly involved in the regulation of the *CD36*, *PPARG*, *SCD1*, *ACSL1*, and *PPARA* genes, while miR-497/*LATS1* is not only involved in the regulation of *CD36* and *PPARG* but also in the expression of the *GPAM*, *ACLY*, *HSL*, *XDH*, *ELOVL6*, *SLC27A6*, and *PLIN3* genes. Finally, although both miR-497/*LATS2* and miR-497/*LATS1* are involved in triglyceride and cholesterol regulation, miR-497/*LATS1* regulates cholesterol at a higher level than miR-497/*LATS2* regulates cholesterol. Thus, many outstanding questions remain to be answered, and more meticulous studies are needed to accurately define *LATS1* and *LATS2*′s shared versus distinct functions. Understanding advances in LATS signaling may help solve basic science mysteries. Moreover, deciphering the details of LATS-mediated tumor suppression will hopefully elucidate opportunities to improve the early detection, prognosis, and treatment of cancer.

## 5. Conclusions

Our data identified for the first time that miRNA-497/LATS1 regulates PUFA metabolism in bMECs, which is a potential factor for increasing the PUFA content in milk.

## Figures and Tables

**Figure 1 genes-14-01224-f001:**
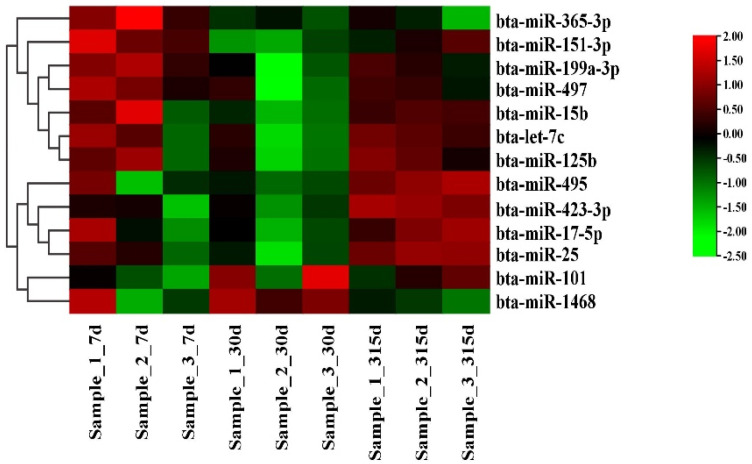
Differential expression of miRNAs in early, peak, and late stages.

**Figure 2 genes-14-01224-f002:**
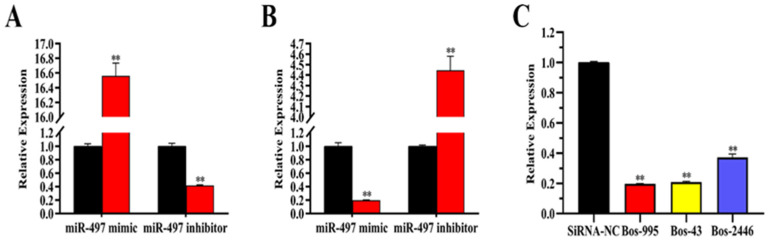
Transfection efficiency of miR-497 and siRNA-*LATS1*. (**A**) Expression level of miR-497. Red bars: the miR-497 mimic or inhibitor; black bars: negative controls. (**B**) mRNA abundance of LATS1 in miR-497 mimic and miR-497 inhibitor groups. (**C**) mRNA abundance of LATS1. Values are presented as means ± standard deviations, ** *p* < 0.01.

**Figure 3 genes-14-01224-f003:**
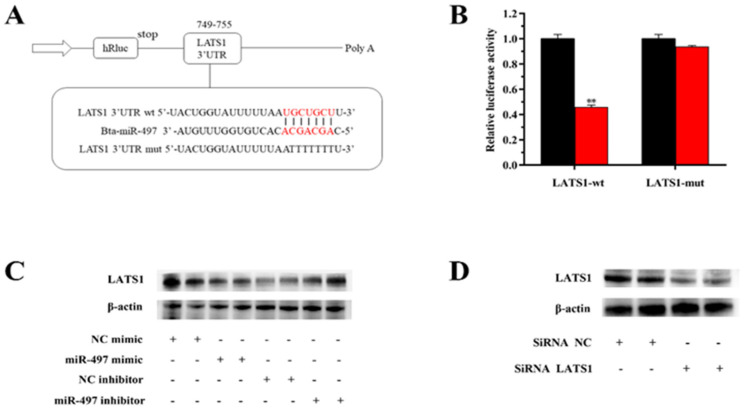
MiRNA-497 specifically targets *LATS1*. (**A**) Target site of miR-497 in the *LATS1* 3′-UTR. (**B**) Luciferase reporter assay. wt: Luc reporter vector in the *LATS1* 3′-UTR; mut: Luc reporter vector with a mutation of the miR-497 site in the *LATS1* 3′-UTR. (**C**) Protein abundance of LATS1. Black bars represent the NC; and red bars represent the miR-497 mimic or inhibitor. (**D**) Protein expression level of LATS1. Protein level of LATS1 after transfection with SiRNA-LATS1 (60 nM) or SiRNA-NC (60 nM). Values are presented as means ± standard deviations, ** *p* < 0.01.

**Figure 4 genes-14-01224-f004:**
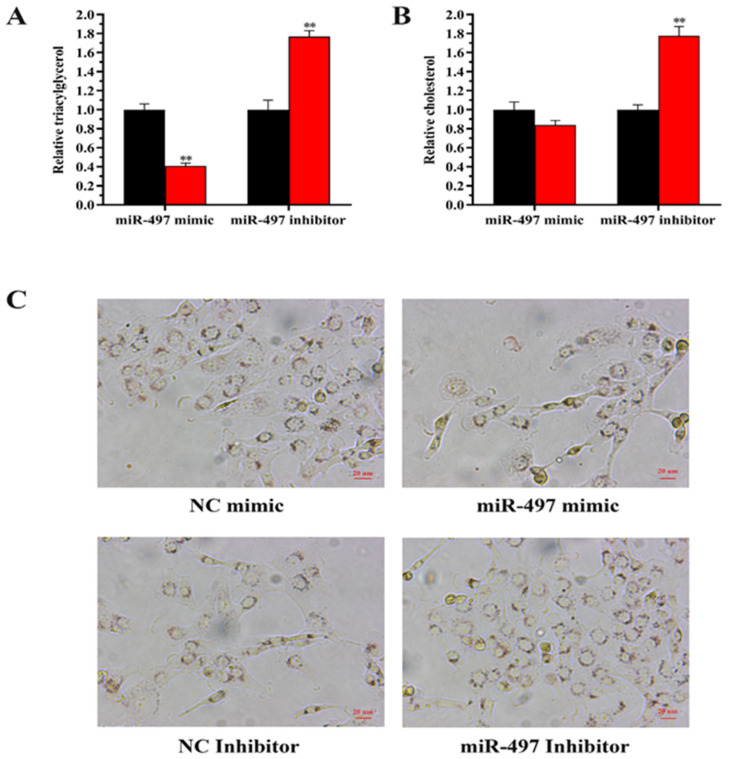
miR-497 functional evaluation. (**A**,**B**) Relative TAG and cholesterol levels. Red bars: the miR-497 mimic or inhibitor; black bars: the negative controls. (**C**) Lipid droplets of BMECs after transfection with miR-497. Values are displayed as the means ± standard deviations (*n* = 6); ** *p* < 0.01.

**Figure 5 genes-14-01224-f005:**
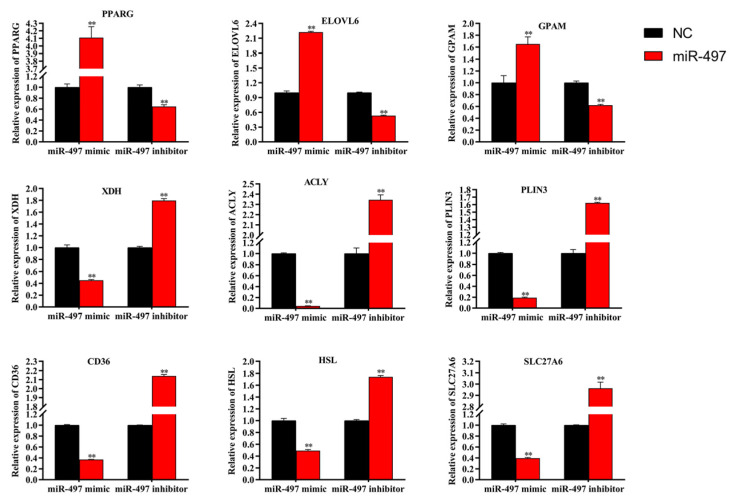
The mRNA expression level. Red bars: the miR-497 mimic or inhibitor. Black bars: the negative controls. All experiments were repeated. Each experiment was repeated three times. Values are displayed as the means ± standard deviations (*n* = 6); ** *p* < 0.01.

**Figure 6 genes-14-01224-f006:**
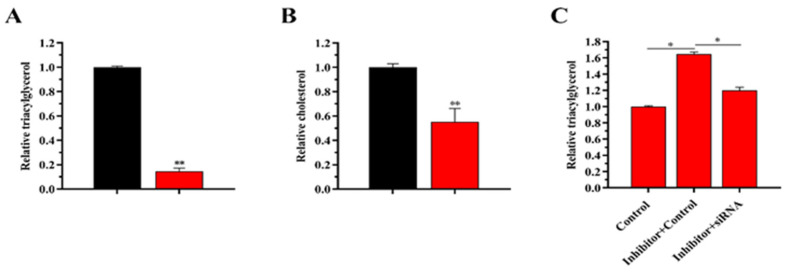
Functional evaluation of *LATS1*. (**A**) Relative TAG levels. Red bars: siRNA-*LATS1*; black bars: the negative controls. (**B**) Relative cholesterol levels. Red bars: siRNA-*LATS1*. Black bars: the negative controls. (**C**) TAG levels; values are presented as the means ± standard deviations; * *p* < 0.05; ** *p* < 0.01.

**Table 1 genes-14-01224-t001:** MiR-497 and siRNA-*LATS1* biosynthetic sequences.

Items	5′-3′	3′-5′
MiR-497 Mimic	CAGCAGCACACUGUGGUUUGUA	CAAACCACAGUGUGCUGCUGUU
MiR-497 Inhibitor	UACAAACCACAGUGUGCUGCUG	
Negative Control	UUCUCCGAACGUGUCACGUTT	ACGUGACACGUUCGGAGAATT
Inhibitor NC	CAGUACUUUUGUGUAGUACAA	
*LATS1*-bos-43	CCAGAAGGAUAUAGACAAATT	UUUGUCUAUAUCCUUCUGGTT
*LATS1*-bos-995	GCAGACAGCCAAUCAUCAUTT	AUGAUGAUUGGCUGUCUGCTT
*LATS1*-bos-2446	GCAGUCGAAAGUGUUCAUATT	UAUGAACACUUUCGACUGCTT

**Table 2 genes-14-01224-t002:** Primer sequences for quantitative fluorescence PCR.

Items	Forward Primer	Reverse Primer
miR-497	5′-GTGCAGGGTCCGAGGT-3′	5′-TAGCCTGCAGCACACTGTGGT-3′
U6	5′-GCTTCGGCAGCACATATACTAAAAT-3′	5′-CGCTTCACGAATTTGCGTGTCAT-3′
*GAPDH*	5′-GTCGATGGCTAGTCGTAGCATCGAT-3′	5′-TGCTAGCTGGCATGCCCGATCGATC-3′
*LATS1*	5′-TCTTTGGTTGGGACTCCTAAT-3′	5′-TTCTTGCCTAAGCGATCTTCT-3′
*GPAM*	5′-ATTGACCCTTGGCACGATAG-3′	5′-AACAGCACCTTCCCACAAAG-3′
*HSL*	5′-GGGAGCACTACAAACGCAACG-3′	5′-TGAATGATCCGCTCAAACTCG-3′
*CD36*	5′-GTACAGATGCAGCCTCATTTCC-3′	5′-TGGACCTGCAAATATCAGAGGA-3′
*ELOVL6*	5′-GGAAGCCTTTAGTGCTCTGGTC-3′	5′-ATTGTATCTCCTAGTTCGGGTGC-3′
*PLIN3*	5′-GGTGGAGGGTCAGGAGAAA-3′	5′-TCACGGAACATGGCGAGT-3′
*PPARG*	5′-CCTTCACCACCGTTGACTTCT-3′	5′-GATACAGGCTCCACTTTGATTGC-3′
*SLC27A6*	5′-CAACTTGCTCATAAACTTTTTCCAAG-3′	5′-TGGTGTGGTTGTGCCAGGT-3′
*XDH*	5′-GATCATCCACTTTTCTGCCAATG-3′	5′-CCTCGTCTTGGTGCTTCCAA-3′
*ACLY*	5′-TTACCCAGAGGAAGCCTAC-3′	5′-AGGATCTTGCCATCTGGGTGC-3′

**Table 3 genes-14-01224-t003:** Effect of miR-497 mimic on fatty acid composition in bMECs.

Fatty Acid (%)	Treatment		*p* Value
	miR-497 NC-Mimic	miR-497 Mimic	
C9:0	7.38 ± 0.25	3.45 ± 0.03	<0.01
C12:0	0.65 ± 0.04	0.97 ± 0.02	<0.01
C14:0	6.07 ± 0.06	8.89 ± 0.06	<0.01
C15:0	0.93 ± 0.03	0.60 ± 0.02	<0.01
C16:0	34.88 ± 0.07	40.67 ± 0.26	<0.01
C16:1	2.73 ± 0.11	1.90 ± 0.04	<0.01
C17:0	1.42 ± 0.08	1.07 ± 0.13	0.03
C17:1	0.39 ± 0.02	0.25 ± 0.02	<0.01
C18:0	23.54 ± 0.10	26.65 ± 0.18	<0.01
9Z C18:1	12.26 ± 0.04	8.21 ± 0.25	<0.01
9E C18:1	5.59 ± 0.14	3.56 ± 0.10	<0.01
9,12(Z,Z)C18:2	1.92 ± 0.05	0.95 ± 0.04	<0.01
C20:0	0.35 ± 0.03	0.67 ± 0.02	<0.01
cis-11C20:1	0.39 ± 0.01	0.30 ± 0.02	<0.01
8,11,14-C20:3	0.33 ± 0.03	0.22 ± 0.03	0.02
5,8,11,14-C20:4(all Z)	0.58 ± 0.04	0.75 ± 0.04	0.01
5,8,11,14,17C20:5(EPA)	0.30 ± 0.00	0.26 ± 0.00	<0.01
4,7,10,13,16,19-C22:6(DHA)	0.28 ± 0.03	0.64 ± 0.04	<0.01

**Table 4 genes-14-01224-t004:** Effects of miR-497 inhibitor on intracellular fatty acid composition in bMECs.

Fatty Acid (%)	Treatment		*p* Value
	NC-Inhibitor	MiR-497 Inhibitor	
C9:0	2.86 ± 0.05	3.87 ± 0.05	<0.01
C12:0	1.67 ± 0.04	0.49 ± 0.03	<0.01
C14:0	9.09 ± 0.09	6.08 ± 0.06	<0.01
C15:0	0.86 ± 0.03	1.57 ± 0.03	<0.01
C16:0	36.42 ± 0.41	31.58 ± 0.30	<0.01
C16:1	3.62 ± 0.06	5.03 ± 0.03	<0.01
C17:0	0.74 ± 0.02	1.89 ± 0.04	<0.01
C17:1	0.33 ± 0.02	0.40 ± 0.02	0.02
C18:0	27.37 ± 0.33	24.11 ± 0.23	<0.01
9Z C18:1	6.92 ± 0.04	11.15 ± 0.26	<0.01
9E C18:1	4.00 ± 0.14	6.62 ± 0.08	<0.01
9,12(Z,Z)C18:2	1.86 ± 0.04	3.57 ± 0.20	<0.01
C20:0	0.61 ± 0.02	0.42 ± 0.01	<0.01
cis-11C20:1	0.33 ± 0.01	0.40 ± 0.02	<0.01
8,11,14-C20:3	0.30 ± 0.04	0.43 ± 0.02	0.02
5,8,11,14-C20:4(all Z)	1.84 ± 0.05	1.26 ± 0.11	<0.01
5,8,11,14,17C20:5(EPA)	0.33 ± 0.04	0.44 ± 0.03	0.04
4,7,10,13,16,19-C22:6(DHA)	0.83 ± 0.02	0.69 ± 0.02	<0.01

**Table 5 genes-14-01224-t005:** Effects of siRNA-*LATS1* on fatty acid composition in bMECs.

Fatty Acid (%)	Treatment		*p* Value
	SiRNA-NC	SiRNA-*LATS1*	
C9:0	8.38 ± 0.25	2.44 ± 0.04	<0.01
C12:0	0.55 ± 0.06	1.57 ± 0.07	<0.01
C14:0	6.65 ± 0.11	6.89 ± 0.06	<0.05
C15:0	0.79 ± 0.03	0.64 ± 0.02	<0.01
C16:0	22.79 ± 0.07	30.27 ± 0.16	<0.01
C16:1	7.22 ± 0.14	3.47 ± 0.06	<0.01
C17:0	1.85 ± 0.10	0.95 ± 0.07	<0.01
C17:1	0.37 ± 0.02	0.24 ± 0.01	<0.01
C18:0	21.42 ± 0.46	32.59 ± 0.16	<0.01
9Z C18:1	16.77 ± 0.07	8.30 ± 0.12	<0.01
9E C18:1	7.26 ± 0.18	5.06 ± 0.02	<0.01
9,12(Z,Z)C18:2	3.25 ± 0.12	1.88 ± 0.06	<0.01
C20:0	0.79 ± 0.06	2.48 ± 0.07	<0.01
cis-11C20:1	0.38 ± 0.02	0.31 ± 0.02	0.03
8,11,14-C20:3	0.35 ± 0.01	0.29 ± 0.01	<0.01
5,8,11,14-C20:4(all Z)	0.61 ± 0.07	0.95 ± 0.03	<0.01
5,8,11,14,17C20:5(EPA)	0.32 ± 0.02	0.26 ± 0.02	0.03
4,7,10,13,16,19-C22:6(DHA)	0.25 ± 0.03	1.42 ± 0.02	<0.01

## Data Availability

The datasets generated and analyzed in this are available at SRR12149782 (https://www.ncbi.nlm.nih.gov/sra/?term=SRR12149782, accessed on 10 July 2022), SRR12149781 (https://www.ncbi.nlm.nih.gov/sra/?term=SRR12149781, accessed on 10 July 2020) and SRR12149843 (https://www.ncbi.nlm.nih.gov/sra/?term=SRR12149843, accessed on 10 July 2020).

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
