# Peer review of "miR-497 Regulates LATS1 through the PPARG Pathway to Participate in Fatty Acid Synthesis in Bovine Mammary Epithelial Cells"

_genes, 2023, doi:10.3390/genes14061224_

Round 1

Reviewer 1 Report

The significant value of the work is the identified potential factor influencing the polyunsaturated fatty acids content in milk.  The authors offer basis to prove that miRNA-497 regulates polyunsaturated fatty acids metabolism by targeting LATS1. This study is important and interesting in terms of in vivo assessment of expression at different stages of lactation in the same animals
However, some additions are required: it is necessary to specify
- The breed name of the investigated animals
- the productivity of the first two lactations.
Because it influences on the result of the experiment.
The authors are also recommended to expand conclusions by adding the specific result.

Author Response

Dear reviewer, thank you for your careful review, Please see the attachment. 

Reviewer 2 Report

The article is interesting but from a methodological point of view fallacious. The authors recently published an article (DOI: 10.1039/d0fo00952k) on the role of miR-491 in the regulation of LATS2 and the consequent effects on bovine mammary gland lipid metabolism (the same model and experimental approach). The writing of the text, tables and figures are also very similar. The reviewer is puzzled not to have read any comments on this. Well, further investigation must be performed to discriminate the effect of miR-497 on LATS1 and LATS2.

The relative effect of miR-497 on the expression/function of LATS1 and LATS2 needs to be inspected.

Lines 58-78 are largely unnecessary.

Lines 104-106, introduce relative references and detailed methods.

Lines 117-121, comments should be moved to the results or discussion section.

Line 130…, indicate distributors for cell and chemicals, indicate transfection conditions, report abbreviations.

Lines 138…., this paragraph sounds like a datasheet, please rewrite.

Lines 145-146, Methods missing.

Lines 149-150, rephrase.

Line 151, reference.

Table 2, correct U6 forward primer.

Lines 164-165, rephrase.

Lines 178-179, rephrase, include model and conditions.

Line 181, fluorescent?

Line 184, means? What do you mean?

Paragraph 3.1, please uniform number format along this text.

Table 3 and 4. unnecessary move to supplementary material; correct “Alighned reads?

Line 202, “common differential expression”, what do you mean?

Line 227-228, to which subset are referred the “*”?

Line 238, please write in methods.

Line 246, please write in methods.

Line 250/251, please rephrase.

Line 264-265, please rephrase.

Line 280, the reported data is the average of what?

Line 297-298, please check the English.

English needs to be greatly improved. There are numerous writing errors.

Author Response

Dear reviewer,

Thank you for your careful review, Please see the attachment, thanks!

Round 2

Reviewer 2 Report

The same authors have previously characterized how "miR-497 regulates fatty acid synthesis through LATS2 in bovine mammary epithelial cells" and in this paper how "miR-497 regulates LATS1 through the PPARG pathway to participate in fatty acid synthesis in bovine mammary epithelial cells." The topic is the same, but again the authors did not put forward hypotheses and comments on how miR-497 regulates fatty acid synthesis through both LATS2 and LATS1: the authors need to elaborate further to distinguish the effect of miR-497 on LATS1 and LATS2.

needs to be improved

Author Response

Dear reviewer,

Please see the attachment, thanks!
